# Stability of Wafer-Scale Thin Films of Vertically Aligned Hexagonal BN Nanosheets Exposed to High-Energy Ions and Reactive Atomic Oxygen

**DOI:** 10.3390/nano12213876

**Published:** 2022-11-02

**Authors:** Shiyong Huang, Zhi Kai Ng, Hongling Li, Apoorva Chaturvedi, Jian Wei Mark Lim, Roland Yingjie Tay, Edwin Hang Tong Teo, Shuyan Xu, Kostya (Ken) Ostrikov, Siu Hon Tsang

**Affiliations:** 1Temasek Laboratories@NTU, 50 Nanyang Drive, Singapore 637553, Singapore; 2School of Electrical and Electronic Engineering, Nanyang Technological University, 50 Nanyang Avenue, Singapore 639798, Singapore; 3School of Materials Science and Engineering, Nanyang Technological University, Singapore 639798, Singapore; 4Plasma Sources and Applications Center, National Institute of Education, Nanyang Technological University, 1 Nanyang Walk, Singapore 637616, Singapore; 5School of Chemistry and Physics and Centre for Materials Science, Queensland University of Technology (QUT), Brisbane, QLD 4000, Australia

**Keywords:** inductively coupled plasmas, chemical vapor deposition, boron nitride, protective layer, ion bombardment, atomic oxygen

## Abstract

Stability of advanced functional materials subjected to extreme conditions involving ion bombardment, radiation, or reactive chemicals is crucial for diverse applications. Here we demonstrate the excellent stability of wafer-scale thin films of vertically aligned hexagonal BN nanosheets (hBNNS) exposed to high-energy ions and reactive atomic oxygen representative of extreme conditions in space exploration and other applications. The hBNNS are fabricated catalyst-free on wafer-scale silicon, stainless steel, copper and glass panels at a lower temperature of 400 °C by inductively coupled plasma (ICP) assisted chemical vapor deposition (CVD) and subsequently characterized. The resistance of BNNS to high-energy ions was tested by immersing the samples into the plasma plume at the anode of a 150 W Hall Effect Thruster with BNNS films facing Xenon ions, revealing that the etching rate of BNNS is 20 times less than for a single-crystalline silicon wafer. Additionally, using O_2_/Ar/H_2_ plasmas to simulate the low Earth orbit (LEO) environment, it is demonstrated that the simulated plasma had very weak influence on the hBNNS surface structure and thickness. These results validate the strong potential of BNNS films for applications as protective, thermally conductive and insulating layers for spacecrafts, electric plasma satellite thrusters and semiconductor optoelectronic devices.

## 1. Introduction

Space missions typically involve harsh environmental conditions, such as ionizing electromagnetic radiation, large and frequent temperature fluctuations, high vacuum and space debris. Conditions such as altitude and inclination also vary depending on the spacecraft’s orbit. In low Earth orbit (LEO) with altitudes of 200–700 km, spacecrafts are also exposed to reactive chemical hazards such as atomic oxygen (AO) species [1]. AO is generated by photodissociation of molecular oxygen in a vacuum environment and irradiation of UV light. AO can cause corrosion and other forms of damage to the spacecrafts, severely limiting its operational lifespan [1,2,3].

Ionizing particles can also come from the spacecraft itself. Ion propulsion engines fire ionized Xe particles to propel the spacecraft forward. It is known for having a long lasting acceleration ability and has attracted the attention of researchers from space agencies such as JAXA and NASA [4,5]. However, one of the issues of ion propulsion engines is the erosion of the engine parts after being constantly bombarded by Xe ions over a long period of time. Xe ions have shown to be able to erode thruster surfaces and expose internal parts, shortening the lifespan [4]. Materials that are resistant to Xe ion bombardment can potentially extend the operation lifetime of ion propulsion engines for deep space missions. While there are a few preliminary reports [6,7], this area of research remains largely unexplored.

In this context, the degradation effects of the space environment on materials have been a topical area of recent studies. For the outer layer of a spacecraft, protective coatings such as $Al_{2}O_{3}$, $SiO_{2}$, $SnO_{2}$ or ITO [8,9,10,11] are commonly used to reduce the AO erosion. To protect the surfaces of optical and optoelectrical devices on a spacecraft, specially designed transparent thin films are required. Hexagonal boron nitride (hBN) has the required properties, as it has strong covalent bonds between the boron and nitrogen atoms. Boron atoms are also naturally attracted to nitrogen atoms due to their higher electronegativity, which ensures the observed high stability. hBN also stands out because of its large bandgap of 6.08 eV [12], which makes it transparent to visible light and UV rays. This implies that hBN films can be used to protect surfaces where we need light to pass through, such as glass or lenses of space equipment.

There are two general routes for synthesis of hBN: top-down (from bulk BN) or bottom-up (from B- and N-precursors). Monolayer and few-layer hBN flakes can be exfoliated from bulk hBN crystals either through mechanical cleavage or a chemical-solution-derived method [13,14,15,16,17]. However, further applications of hBN flakes exfoliated using these top-down techniques are limited by the flake size and homogeneity over large areas. Advances in synthesis techniques have improved the material quality of grown hBN [18] and enabled the growth of single-layer hBN. Chemical techniques also offer significant advantages for synthesizing large-area hBN films. Recently, mono- and few-layer BN nanosheets (BNNSs) have been synthesized via microwave plasma chemical vapor deposition (MPCVD) [19] and thermal chemical vapor deposition (CVD) [20]. While these methods have the potential to grow large area, homogeneous BN layers, they require a high temperature to either directly grow the BN, or to anneal the precursors into the BN. A lower deposition temperature that still allows uniform deposition of the BN would be a promising approach to protect parts from the harsh space environment.

In this study, we explored the fabrication of thin films of vertically aligned hexagonal BN nanosheets (hBNNSs) using an Inductively Coupled Plasma (ICP) CVD method directly conducted on wafer-scale (4 in) substrates of different materials at a low substrate temperature of 400 °C. The thickness and alignment of the hBNNSs were controlled by optimizing the deposition conditions. We also studied the etching effect of Xe ions on CVD grown BNNSs and quantified the etching rate for the given parameters. It was proven that the hBNNS thin films could resist the bombardment of high density and high energy Xe ions generated by an ion thruster for at least 80 h. The synthesized BNNSs can also withstand AO in the ICP-simulated environment for at least 40 h. The morphology and thickness of the BNNSs after Xe ion bombardment and AO exposure were analyzed and it was concluded from this that the synthesized BNNSs are stable and resistant to high energy ions and AO exposure.

## 2. Methods

### 2.1. Sample Preparation and Characterization Methods

All the fabrication experiments in this study were conducted using the inductively coupled plasma (ICP) CVD apparatus [21,22,23]. Briefly, the wafers used were first cleaned using the general Radio Corporation of America (RCA) silicon wafer cleaning process. The other substrates were sequentially cleaned via ultrasonication with by carbon tetrachloride, acetone, alcohol and deionized water. The substrates were soaked in each solution and sonicated for 5 min to ensure the substrates were clean. The substrates were loaded in the ICP CVD chamber and the temperature was slowly raised to 400 °C over the course of 90 min in a low-vacuum environment. Ar and $H_{2}$ gases were then introduced into the chamber for 10 min to clean the substrate. The ICP was then turned on using the different specified powers and the temperature was allowed to stabilize. Subsequently, $B_{2}H_{6}$ and $N_{2}$ gas were introduced into the system as the boron and nitrogen precursor. It is to be noted that BNNSs of up to 1 µm were successfully deposited. Characterization methods: Transmission Electron Microscopy (TEM) was conducted using JEOL 2010F. A JEOL JSM-6700 FE-SEM high resolution scanning electron microscope was used to take SEM images of the surface and measure the cross-section thickness of the films and substrates. Raman spectroscopy was carried out using Renishaw inVia Raman with 514 nm laser. Fourier transform infrared (FTIR) spectroscopy was conducted using a Bruker VERTEXV 80 V. X-ray Photoelectron Spectroscopy (XPS) was conducted using the Kratos Analytical Axis Supra X-Ray Photoelectron Spectrometer. Xenon ions were produced using a custom-made ion propulsion device. Xe gas was injected at the cathode and anode at 0.6 and 6 sccm, respectively. The inner magnetic coil, outer magnetic coil and anode were connected in series, with the voltage set at 270 V and the current at 0.5 A. The total DC power supply of thruster was about 150 W. The average ion density and average ion energy of the Xe plume is 1.6–6.4 × 10^16^ cm^−2^. This density and average ion energy are expected from an ion thruster and hence would be sufficient to simulate Xe bombardment from the ion thruster. The AO environment is simulated in the ICP chamber. The substrate temperature was set at 100 °C, which is the average temperature of the LEO environment. Using a mass flow of $O_{2}$, Ar and $H_{2}$ at 6:4:20 (sccm) and ICP power of 2.0 kW, AO can be generated. This approach is similar to the approach adopted by other groups [24]. A higher $O_{2}$ flow and microwave power were used to ensure that there is sufficient AO generated to simulate the LEO environment. Thickness of the BNNS layer was measured before and after being left in the environment. All etching studies were repeated at least 3 times to ensure reproducibility.

### 2.2. Characterization of BNNSs

In order to obtain uniform BNNS films, the wafers were placed on a rotating platform above the heater, with the rotating speed easily controlled by an electric motor. Additionally, the substrates were cleaned with a general Radio Corporation of America (RCA) silicon wafer clean process; the other kinds of substrates were cleaned by RS PRO ultrasonic cleaner with the order of carbon tetrachloride, acetone, alcohol and deionized water agent for about 5 min. Before the deposition of hBNNSs, the substrates were still etched for about 10 min by the $H_{2}$ and Ar plasma; then $B_{2}H_{6}$ was introduced to the chamber; the hBNNS thin films were produced according to the setting parameters.

## 3. Results and Discussion

### 3.1. SEM Imaging of the hBNNS Thin Films

Figure 1 shows (Scanning Electron Microscope JEOL 7600) the entire process of hBNNS grown on a silicon substrate (with similar observations for other substrates), keeping the experimental conditions constant (gas flow ratios $N_{2}$:$B_{2}H_{6}$:Ar:$H_{2}$ = 3:0.5:1.2:14, substrate temperature: T = 400 °C and ICP discharge input power PICP = 2.0 kW). Initially, a polycrystalline transition BN layer was formed over the entire surface, with a thickness of approximately 15 nm and an extremely smooth appearance of the whole surface (Figure 1a). As the deposition process continued, BN nanoparticles were formed on the surface (Figure 1b), which could act as the seed layer for the large-area BNNSs. Subsequently, individual separated small-sized BN nanoparticles formed on the substrate, increasing in size with longer deposition time (Figure 1c). During the final stage, the BNNSs merged with one another and were aligned vertically to the substrate surface (Figure 1d). When the deposition duration was extended beyond 90 min, the surface morphology remained the same (Figure 2a).

To study the effects of different substrates on the formation of hBNNSs, we placed the glass, stainless steel and silicon substrates on the holder at the same time, with the deposition time increased to 300 min, keeping other parameters the same as in Figure 1. As demonstrated in Figure 2, hBNNSs could be grown on stainless steel (Figure 2b) and glass (Figure 2c) substrates, with the surface morphologies and thickness of the transition BN layer being largely similar. From Figure 2d, as the deposition process continues, polycrystalline BN material eventually fills the entire space among the hBNNSs.

The density and size of hBNNSs could be effectively controlled by the deposition parameters, such as mass flow ratio of $B_{2}H_{6}$ and $H_{2}$, power supply of the RF ICP, substrate temperature and the total pressure of the deposition chamber. Our experimental results show that the ICP power and substrate temperature had more influence on the formation of hBNNS when the ICP power was below 1.2 kW. When the temperature was below 200 °C, no hBNNS was formed on each substrate surface.

### 3.2. Structural Properties of hBNNS Thin Films

Raman spectroscopy was further conducted to characterize the synthesized hBNNS structures. Figure 3a shows the typical Raman spectra of the samples of Figure 2, with the characteristic peak measured at 1368 cm^−1^, similar to what many other groups have reported. [14,25]. The full width at half maximum (FWHM) of the samples was 28, 34 and 35 cm^−1^, respectively. This Raman spectra revealed that the hBNNSs synthesized under the above process were of high purity. Meanwhile, it was proven that the type of the substrate has some influence on the structure of the hBNNS. Figure 3b shows the Raman spectra of a series of hBNNS thin films deposited on a silicon substrate at 380 °C with different ICP power supply ranging from 1.5 to 2.5 kW, keeping other fabrication parameters the same as those in Figure 2. Even when the ICP power supply was only 1.5 kW, a thin film of hBNNS was still produced on the substrate (shown in Figure 4a), although the density and thickness were less than those of the sample shown in Figure 2a.

Fourier transform infrared (FTIR) spectroscopy (Bruker VERTEX 80 V) was conducted to confirm the phase purity and chemical structure of the hBNNS. As shown in Figure 4b, two distinct peaks at 1392 cm^−1^ and 804 cm^−1^ were observed. These two peaks correspond to the in-plane stretching of B–N ($E_{1u}$) and the through-plane bending vibration of B–N–B ($A_{2u}$), respectively. [14,26,27,28]. The absorption peak at 1065 cm^−1^ corresponds to the TO phonon mode, originating from the padding of the BN material [29].

Transmission electron microscopy (TEM) was conducted with a JEOL 2010F to characterize the structure of the hBNNS, with Figure 5 showing the TEM images of the sample. A small piece (5 × 5 mm^−2^) was obtained from the sample in Figure 2a and was transferred to a little beaker (25 mL), covered with purified ethanol with ultrasonic method (10 min). Thereafter, small pieces of BN sheets (micrometer scale) were moved to carbon coated copper nets of the TEM substrate holder. In the high-resolution TEM Figure 5b, parallel line features can be observed along the edge of the film under high magnification [14,25,26,29]. The interlayer distance measured from our TEM image is 0.34 ± 0.02 nm, consistent with the reported value for the structure of the hBN [25]. The hexagonal lattice with a spacing of 0.21 ± 0.02 nm is also shown in Figure 5b. The values are consistent with the reported value for bulk hBN [25]. These TEM characterizations of the hBN film confirmed the crystalline nature of the synthesized films.

### 3.3. hBNNS Thin Films Etching by Higher Density and Energy Xe Ions

During the stable state, the mass flow of Xe gas at the cathode and anode were 0.6 and 6 sccm (standard cubic cm per min) [24,30]. The inner, outer magnetic coil and anode were connected in series, with the voltage set at 270 V and the current at 0.5 A. The total DC power supply of thruster was about 150 W. We counted that the Xe ion density and average energy in the plume at 30 cm location were 1.6–6.4 × 10^16^ cm^−2^ and 135 eV, respectively. To test the protective function of BNNS thin films, one small piece of BNNS sample (cut from the sample in Figure 2a) and one piece of single-crystalline silicon were put at the center of the plasma plume at the anode of a 150 W Hall effect thruster constructed by our group [30,31] (Figure 6a). The centers of the square sample and the anode were aligned, kept at a distance of 30 cm apart with the BNNS films directly facing the Xenon ions. During the ion etching process, all the parameters of the Hall effect thruster were maintained at the same values. Figure 6a–d present SEM images of the surface morphologies of the BNNS films at various time intervals throughout the ion etching process. The surface roughness of the material decreased from R_*a*_:6.1 nm and R_*q*_:8.0 nm before and R_*a*_:4.2 nm and R_*q*_:5.0 nm after 80 h of Xe ion bombardment. The reduction in surface roughness is also observed when the SEM images between Figure 2c and Figure 6c are compared.

One possible explanation for the differences in appearance of the surface across time could be due to the polycrystalline BN layer within the BNNS film being displaced by the impact of high-energy Xe ions. Since boron and nitrogen atoms of the vertical BNNS were bound by strong covalent bonds, they can withstand the impact of these ions at high densities and speeds. However, as the process continues, the vertical BNNSs were increasingly bent at an angle, eventually forming a protective layer over the surface. This explanation could be proven by the residue thicknesses shown in Figure 6b,c, which decreased from an initial value of 325 nm to 304, 291, 284 and finally 279 nm, in intervals of 20 h. This gives an average etching rate of 0.575 nm/h. Since a significant portion of the BNNS layer is still present and no pinholes were observed on the BNNS layer, the underlying substrates are inferred to be unaffected by the etching. Importantly, the etching rate of hBNNS thin films was only 8.2% that of a single crystalline silicon wafer in the same Hall effect thruster anode plasma plume, with the thickness of a single crystalline silicon reduced by approximately 140 nm every 20 h. It is acknowledged that more can be done to quantify the stability of the hBNNS film and this will be done in detail in our subsequent work.

### 3.4. hBNNS Thin Film Etching by Simulated Atomic Oxygen

We used the same ICP apparatus as the deposition of hBNNS to simulate AO on LEO, with mass flow of $O_{2}$, Ar and $H_{2}$ at 6:4:20 (sccm) and ICP power at 2.0 kW. The sample was obtained from the same sample in the thruster etching process above. The substrate temperature was set at 100 °C, which was the average temperature at LEO. To create a simulated AO environment, we used an ICP apparatus, similar to the approach adopted by other groups [24]. The AOs were generated from an oxygen supply while Ar and $H_{2}$ were used as supplementary gases. The surface morphology of the sample after 40 h under ICP plasma exposure is shown in Figure 6d. The thickness of the hBNNS only decreased by 5 nm, showcasing outstanding resistance against AO. Raman spectroscopy was conducted to confirm the quality of the BNNS after AO exposure. Indeed, from Figure 7, there is no observed oxidation of BNNS after 40 h of AO exposure. This is similar to what was observed by other groups when bulk BN is subjected to thermal oxidation [32,33]. The protection mechanisms of BN layers from oxygen have been well studied and have proven to be effective at one to a few layers thick [34,35,36]. Since there is still a thick layer of BNNS present after the AO exposure, the underlying substrate is unaffected.

The surface oxidation states of BNNS thin film before and after AO exposure were also characterized using XPS. A clear B-N and N-B signal at 190.5 eV and 398.2 eV can be detected from the B1s and N1s spectra (Figure 8a,b) before exposure to the AO [37]. After 40 h of exposure, B-O bonds with a characteristic peak of 193.5 eV is found [38]. In the N1s spectra, two peaks are shown at 398.9 eV and 402.0 eV, which correspond to the N-C and N-O bonds, respectively [39]. The formation of the N-C bonds is likely due to the interaction of adventitious carbon with the nitrogen of the BN under the plasma of our simulated AO environment. From the XPS and Raman data, it is evident that the AO indeed interacted with the BNNS thin film. However, as the deposited BNNS is 300 nm, only the surface interacts with the AO, leaving most of the BNNS intact as BN. Therefore, B-O and N-O bonds can be observed on the surface via XPS, but the BN remains largely intact as shown by the Raman spectroscopy in Figure 8.

Furthermore, the electrical resistivity of the BNNS films in our experiment reached values between $10^{12}$ and $10^{14}$ Ωcm, depending on their microstructure. The main factor influencing the resistivity was the amount and nature of the grain boundary phase. The average transmittance from deep UV to IR was larger than 92% [40]. This implies that our hBNNS thin films can potentially be applied to materials like glass and similar surfaces to prevent them from coloration due to AO exposure, while allowing the light to pass through [41]. We acknowledge that more can be done to compare our BNNS film with other similar protection layers. However as the testing parameters and BN thicknesses are vastly different, comparisons made would be ambiguous and not meaningful.

## 4. Conclusions

Vertically aligned hexagonal BNNS thin films were fabricated in a low temperature and catalyst-free process on several kinds of substrates using the ICP CVD technology. Various aspects of the BNNS thin films can be controlled by tuning the synthesis parameters. Such hBNNS films could withstand fluxes of high energy Xe ions and reactive atomic oxygen. The etching rate of high energy Xe ions on the BNNS nanosheet is also quantified. With 1 µm of the synthesized BNNS, we are able to protect the underlying substrate from approximately 2000 h of high energy Xe ions and 8000 h of AO. These findings support the perspectives of applications of hBN films as protective and insulation layers on electric space thrusters and diverse semiconductive devices expected to operate in harsh environments not limited to space exploration.

## Figures and Tables

**Figure 1 nanomaterials-12-03876-f001:**
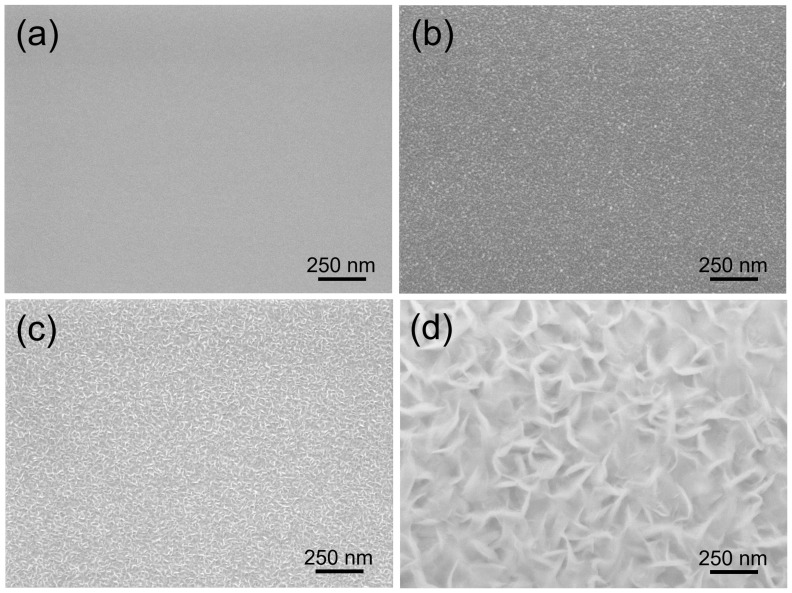
SEM images of BN films deposited on Si wafers under the same conditions, except for varying deposition times of (**a**): 10 min; (**b**) 30 min; (**c**) 60 min; and (**d**) 90 min.

**Figure 2 nanomaterials-12-03876-f002:**
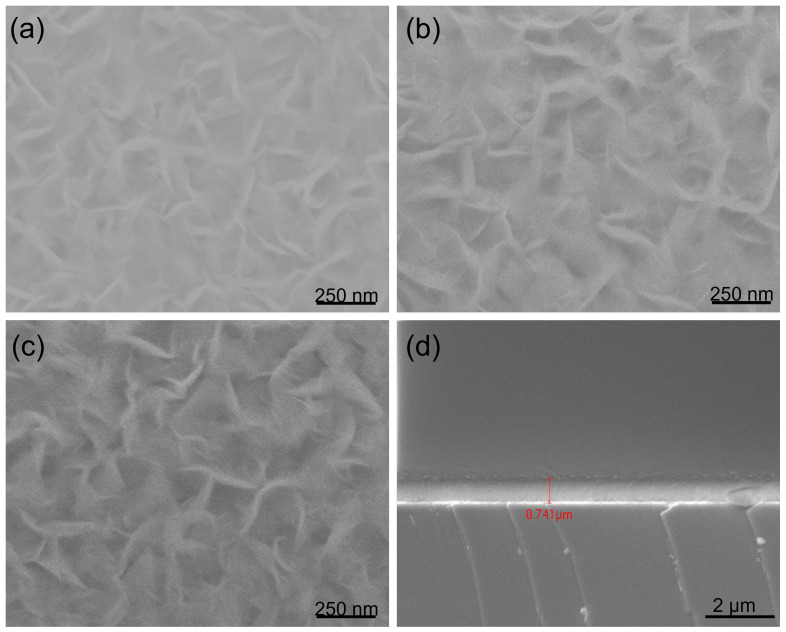
SEM images of BNNS films deposited on (**a**) Si wafer, (**b**) Stainless Steel, (**c**) Glass, substrates respectively under the same conditions as in Figure 1, with only the deposition duration extended from 90 to 300 min. (**d**) A cross-section of (**a**).

**Figure 3 nanomaterials-12-03876-f003:**
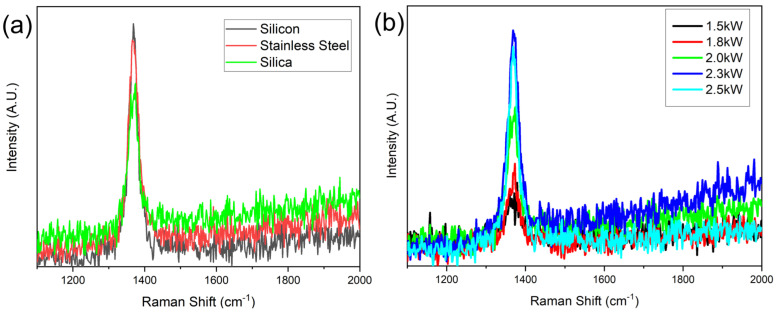
Raman spectra of (**a**) the samples in Figure 2 and (**b**) the samples with different ICP power at 380 °C substrate temperature.

**Figure 4 nanomaterials-12-03876-f004:**
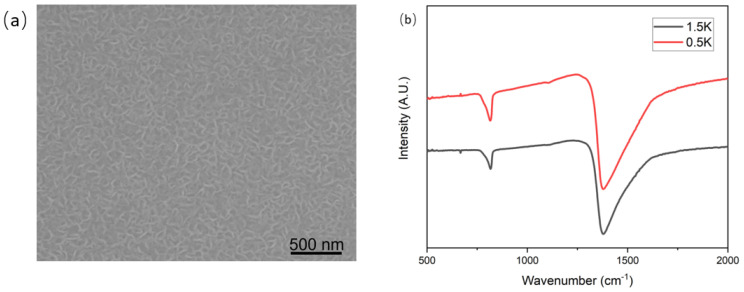
(**a**) SEM image of hBNNS with 1.5 kW ICP power, (**b**) FTIR spectra of BN films deposited on Si wafers with different ICP RF power supply, with other parameters kept the same as those of the sample shown in Figure 2a.

**Figure 5 nanomaterials-12-03876-f005:**
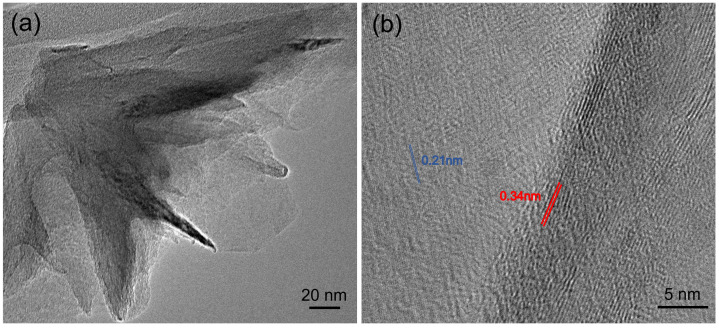
(**a**) Low-resolution TEM image and (**b**) high-resolution TEM image coming from the same sample of Figure 2a.

**Figure 6 nanomaterials-12-03876-f006:**
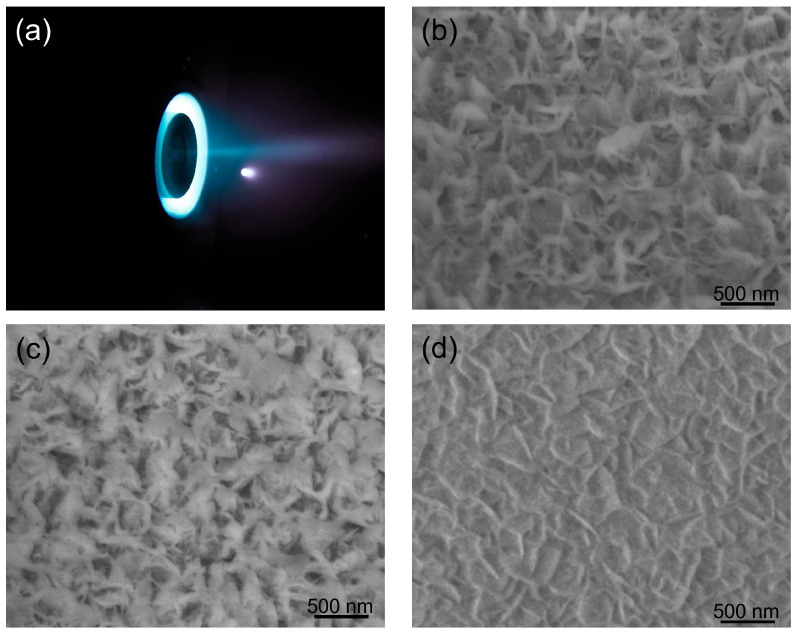
Photograph of the 150 W Hall effect thruster (**a**) during its stable working state and SEM images of BNNS samples with the different etching durations: (**b**): 40 h; (**c**): 80 h; and (**d**): AO etching 40 h.

**Figure 7 nanomaterials-12-03876-f007:**
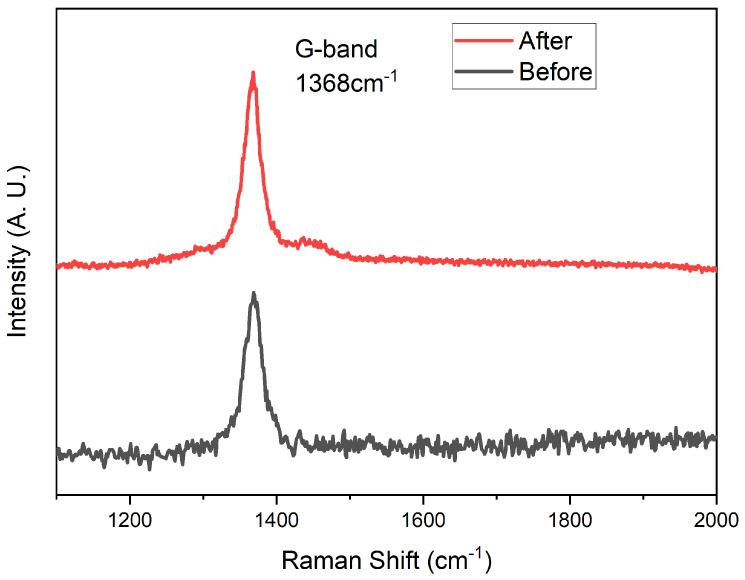
Raman spectroscopy of the BNNS before and after the AO exposure.

**Figure 8 nanomaterials-12-03876-f008:**
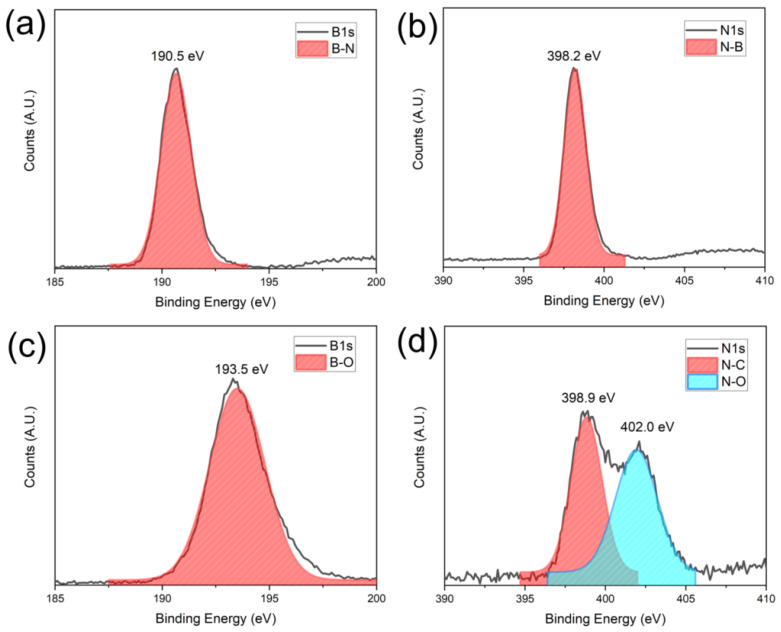
XPS signatures of the BNNS before and after AO exposure: (**a**) B1s and (**b**) N1s signal before exposure and (**c**) B1s and (**d**) N1s after exposure.

## Data Availability

Not applicable.

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
