# Peer review of "Stability of Wafer-Scale Thin Films of Vertically Aligned Hexagonal BN Nanosheets Exposed to High-Energy Ions and Reactive Atomic Oxygen"

_nanomaterials, 2022, doi:10.3390/nano12213876_

Round 1

Reviewer 1 Report

The proposed manuscript is dedicated to the formation and stability evaluation of hexagonal BN nanosheets coatings taking into account the possible application in the aerospace industry. The research topic can be considered as important and the initial results demonstrated by the authors deserve special attention.

Nevertheless, the manuscript in the proposed form is far from the modern requirements for publication from a scientific point of view and representation quality.

1)      Representation issues:

i)        The authors did not provide a detailed description of the procedures for the preparation and instrumental techniques for the characterization. Moreover, the experimental section is missing in the manuscript and relevant supporting information was not provided.

ii)       The results and discussion section are presented poorly. The authors did not provide sufficient discussion of the changing in surface morphology. The manuscript text does not have a clear message for the scientific community.

2)      Scientific issues.

i)        The authors made the in-depth conclusion without sufficient support from experimental data. The analysis of stability cannot be considered as full without careful analysis of surface chemistry.

ii)       The mechanical stability tests should be provided. Moreover, the surface roughness should be determined before and after treatment.

iii)     The quantitative comparison of stability with other materials described in the introduction section should be provided.

Thus, the manuscript required serious corrections and addition of experimental details. The manuscript does not meet the criteria of quality and scientific soundness required for publication in Nanomaterials journal. Nevertheless, it can be accepted after correction and subsequent resubmission. 

Author Response

Reviewer 1 (Reviewer's comments in black, and response is in red)

Comments and Suggestions for Authors

The proposed manuscript is dedicated to the formation and stability evaluation of hexagonal BN nanosheets coatings taking into account the possible application in the aerospace industry. The research topic can be considered as important and the initial results demonstrated by the authors deserve special attention.

Nevertheless, the manuscript in the proposed form is far from the modern requirements for publication from a scientific point of view and representation quality.

1)      Representation issues:

  1. i)        The authors did not provide a detailed description of the procedures for the preparation and instrumental techniques for the characterization. Moreover, the experimental section is missing in the manuscript and relevant supporting information was not provided.

Thank you for your above comments. We have added a dedicated Methods section into the paper following the reviewer’s recommendations on page 2-3. The revised portions of the manuscript are copied below for the convenience of review, and are available in section 2.1

All the fabrication experiments in this study were conducted using the inductively coupled plasma (ICP) apparatus [22– 24]. Briefly, the wafers used were first cleaned using the general Radio Corporation of America (RCA) silicon wafer cleaning process. The other substrates were sequentially cleaned via ultrasonication with by carbon tetrachloride, acetone, alcohol, and deionized water. The substrates were soaked in each solution and sonicated for 5 min to ensure the substrates were clean. The substrates were loaded in the ICP CVD chamber and the temperature was slowly raised to 400°C over the course of 90 min in a low-vacuum environment. Ar and H2 gases were then introduced into the chamber for 10 min to clean the substrate. The ICP is then turned on using the different specified powers, and the temperature was allowed to stabilise. Subsequently, B2H6 and N2 gases were introduced into the system as the boron and nitrogen precursors, respectively.

Characterisation Methods: Transmission Electron Microscopy (TEM) was performed using JEOL 2010F. JEOL JSM-6700 FE-SEM high resolution scanning electron microscope was used to take the SEM images of the surface and measure the cross-section thickness of the films and substrates. Raman spectroscopy relied on the Renishaw inVia Raman spectroscope with the 514
nm laser excitation. Fourier transform infrared (FTIR) spectroscopy was conducted using a Bruker VERTEXV 80V. Xenon ions were produced using a custom-made ion source. Xe gas was injected at the cathode and anode at 0.6 and 6 sccm, respectively. The inner magnetic coil, outer magnetic coil, and anode of the ion source were connected in series, with the voltage set at 270 V,  and the current at 0.5 A. The total DC power supply of the ion source was about 150 W. The average ion density and the average ion energy of the Xe plume is 1.6 – 6.4 x 1016 cm−2. This density and average ion energy is expected from an ion thruster, and hence would be sufficient to simulate Xe bombardment from the ion thruster. Atomic oxygen (AO) environment is simulated in the ICP chamber. The substrate temperature was set at 100°C, which is the average temperature of the LEO environment. Using a flow of mass flow of O2, Ar, and H2 gases at 6 : 4: 20 (sccm) and ICP power of 2.0 kW, AO can be generated. The thickness of the BNNS layer was measured before and after the exposure to the environment.

  1. ii)       The results and discussion section are presented poorly. The authors did not provide sufficient discussion of the changing in surface morphology. The manuscript text does not have a clear message for the scientific community.

Thank you for your comments, we apologize for not explaining it clearly before. We followed the Reviewer’s suggestion and have made substantial changes to the flow of the introduction to explain the the clear message and the importance of our work to the community. Specifically, we want to emphasize that our ICP induced BN coatings can be grown at lower substrate temperatures, while maintaining the quality of the BN grown. We have also conducted and quantified the BN etching rate under Xe ion bombardment and simulated AO environment. We have also provided additional information to prove the quality of the BN after exposing to AO. Please see the introduction and the final part of the paper for more information (Page 1-2 for the introduction, and Page 9 for the additional information of BN quality after AO etching.).

2)      Scientific issues.

  1. i)        The authors made the in-depth conclusion without sufficient support from experimental data. The analysis of stability cannot be considered as full without careful analysis of surface chemistry.

We thank the Reviewer for this important comment. We have followed it and clarified on what we define as “stable” for this work.

We define the stability of the BN layer as the presence and the homogeneity of the BN layer after the bombardment of Xe ions or AO exposure. To that end, we have verified the quality of the BN via SEM, and it is presented in figure 6. Unlike other works where only single or few layers of BN is present, we started with an initial BN thickness of 325 nm. After the Xe+ ion bombardment and AO exposure, the leftover BN thickness is still very significant. We have now added the results of Raman spectroscopy before and after exposure to confirm the quality of BN after AO exposure (please refer to Page 9 for the additional information of BN quality after AO etching.).

Based on our experimental data, we conclude that the BNNS layer is stable. We have added this definition at the end of the introduction on page 2 to clarify. Thank you for your suggestion.

With regards to surface chemistry, there are a few papers that have already reported the protection mechanism of BN from atomic oxygen. The BNNS in this work should also follow the same protection mechanism. We have added and cited these papers on page 9 of the discussion, and copied the references here for the convenience of review.

Yi et al., 2014. Boron nitride nanosheets as oxygen-atom corrosion protective coatings. Applied Physics Letters104(14), p.143101.

Zhang et al., 2017. Preparation of high-content hexagonal boron nitride composite film and characterization of atomic oxygen erosion resistance. Applied Surface Science402, pp.182-191.

Liu et al, 2018. Impermeability of boron nitride defect-sensitive monolayer with atomic-oxygen-healing ability. Ceramics International44(12), pp.13888-13893.

  1. ii)       The mechanical stability tests should be provided. Moreover, the surface roughness should be determined before and after treatment.

Thank you for this valuable comment. We agree that studying the mechanical stability of the BNNS synthesized helps to deepen the understanding of the material. However, based on our current observations, the BN layer is still present on the substrate and of good quality after Xe ion bombardment and AO exposure. Since the main message of our paper is to demonstrate the in-principle resistance of BNNS to the Xe ion and AO exposure, this information may not add substantially to the discussion in this work. Nonetheless, we have added this study into our plans for future work and briefly mentioned on page 9 of the revised manuscript.

The roughness of the BNNS is determined to be Ra: 6.1nm and Rq: 8.0nm before and Ra:4.2nm and Rq:5.0nm after bombardment. Regarding the recommended surface roughness, we have added this information on page 8.

iii)     The quantitative comparison of stability with other materials described in the introduction section should be provided.

Thank you very much for this valuable suggestion. The authors do agree that a quantitative comparison would benefit and strengthen the case for our paper as well. However, simulated AO environment studies has limited papers, so we compared our results with what we were able to find in literature and have cited the relevant paper (page 9 and Reference 25 in the revised manuscript).

We did not observe pinholes that were formed after 40hrs of AO bombardment as seen in Figure 6d. We have also added a Raman study in Figure 7 to prove that the BN quality did not deteriorate after the AO bombardment.

Thus, the manuscript required serious corrections and addition of experimental details. The manuscript does not meet the criteria of quality and scientific soundness required for publication in Nanomaterials journal. Nevertheless, it can be accepted after correction and subsequent resubmission. 

We greatly appreciate the Reviewer for the insightful and constructive suggestions. By addressing all these comments and suggestions, we believe that the paper is much better now. 

Reviewer 2 Report

The paper present characterisation of BN sheets prepared in CVD process for protective layers is space/high attlitude atmosphere. Thus it fits the scope of the journal although use of BN sheet as protective layer has been reported in publication. Thus subject is not highly original. The Author have not fully proved application potencial. The paper has several drawbacks and requires improvement:

1.Material should be compared with existing solutions for this application with set of objective, numerical parameters.

2. Set of required parameters of protective BN sheets should be given. Required and obtained.

3. Aim of the reserach ich not clear. Have the Authors tested that BN sheets would only survive or survive and protect the other material in harsh environment?

4. The Author should compare conditions that they created in lab for tests with real condition with set of objective parametrs in low orbit and prove this way that the their approach (test conditions) is sufficient.  

5. Did the Authors make mapping of the structure (e.g. Raman) before and after the exposition?

6. Did the Authors repeated the experiment and got the same results?

7. Section "Conclusions" is too short and too general. It was not confirmed by numerical parameters that success was achieved.

Author Response

Reviewer 2 (Reviewer's comments in black, responses in red)

Open Review

The paper present characterisation of BN sheets prepared in CVD process for protective layers is space/high attlitude atmosphere. Thus it fits the scope of the journal although use of BN sheet as protective layer has been reported in publication. Thus subject is not highly original.

We thank the Reviewer for this insightful suggestions. Indeed, BN sheets prepared via CVD for protection is an important research field. The BNNS in this paper were synthesized with a modified CVD method, and at a lower temperature. The thickness of the BNNS layer can also be controlled. This can allow more substrates to be coated with different thicknesses of BN for protection purposes. However, the protection ability of this low temperature synthesized BN needs to be quantified. Therefore, our methods and findings are still novel and valuable.

We apologize that this point was not made clear in our manuscript and have reworded the introduction to emphasize the importance of the advances made in our work. Please see page 2 of the revised manuscript.

The Author have not fully proved application potential. The paper has several drawbacks and requires improvement:

1.Material should be compared with existing solutions for this application with set of objective, numerical parameters.

Thank you for the valuable input. In the last section of the study of the effect of AO exposure on BNNS film, we have reported that only 5 nm of BNNS have been lost after 40 hours of AO exposure. It is also well known that BNNS layers are transparent to a wide range of wavelengths with a transmittance of at least 92% over UV to IR waves [see, e.g., Snure, M., et al.2014. Optical characterization of nanocrystalline boron nitride thin films grown by atomic layer deposition. Thin Solid Films, 571, pp.51-55].  

Combining with our previous response related to the advantages of the ICP CVD synthesized BNNS, there is potential for this BNNS to be applied to surfaces where it needs to be protected from AO, while still being transparent to certain wavelengths of light. One such example would be that BNNS can protect the glass from AO, preventing coloration while allowing light to pass through [see e.g., De, et al., 1991. Coloration of glass exposed to atomic oxygen. Journal of materials engineering13(3), pp.213-216].

(For application of BNNS for Xe+ ion protection, please see response from section 2.)

We also agree that the material should be compared with the existing solutions over some parameters to strengthen our case. However, there are not many works that studied the protection ability of materials against Xe+ ions. It is also an ambiguous comparison if we compare other works that use few layered BN and/or AO beam etching because the parameters are vastly different. Therefore, we can only do a basic and simple comparison with other common materials, such as single crystal silicon wafer. The relevant discussion is added on page 10  of the revised manuscript.

  1. Set of required parameters of protective BN sheets should be given. Required and obtained.

Thank you for this comment. As of now, there is no standard that is stated by the community for protective BN layers. However, we can make some inferences based on the recently published work.

According to the relevant recent paper [David et al.., 2020. Ion Propulsion Technology: NASA's Evolutionary Xenon Thruster (NEXT) Development and Long Duration Tests Results and its Applications. In 2020 Advances in Science and Engineering Technology International Conferences (ASET) (pp. 1-5). IEEE.], some parts of the ion thruster engine are subjected to extended lifetime tests upwards of 50000 hours, after which severe etching (or erosion) of the parts were observed. One component called the keeper discharge electrode “eroded enough to expose several internal parts such as the radiation shielding, cathode heater and several others…”

From our results, BNNS etches at a rate of ~ 0.5nm/hr when bombarded by Xe ions. Using the parameters stated in Section 2, we are able to grow BNNS at about 50nm/hr and have managed to grown beyond 1 micrometre thickness. This means that with 1 micrometre thick BNNS film, we can protect the substrate and extend the lifespan of the ion thruster by at least 2000 hours. With only 25 micrometre of our BNNS film, it will be sufficient to protect the equipment for the entire testing timeline.

Therefore, protection from Xe ions is critical for the operation lifespan of the ion thruster, and our BNNS has the potential to significantly extend the operation lifespan of ion thrusters. The relevant discussions has been added on page 1-2 and page 11 of the revised manuscript.

  1. Aim of the research ich not clear. Have the Authors tested that BN sheets would only survive or survive and protect the other material in harsh environment?

Thank you for this important suggestion, we aim to show that our ICP CVD grown BN is able to protect the underlying substrates from Xe ion and AO. We have tested and confirmed that the underlying substrate survives the Xe ion bombardment and the AO bombardment.

We have now also included a Raman analysis of the BNNS before and after AO exposure to confirm the quality of the BN.

The relevant revisions can be found on page 8-10 of the revised manuscript.

  1. The Author should compare conditions that they created in lab for tests with real condition with set of objective parameters in low orbit and prove this way that the their approach (test conditions) is sufficient.

Thank you for this valuable suggestion. For Xe ion bombardment, we used an ion source which could potentially be used in ion thrusters for space propulsion applications in near future. The bombardment was performed is a vacuum chamber to simulate the LEO environment. The authenticity of the Xe ion source and the low vacuum environment should be sufficient to simulate the LEO environment in the lab.

As for AO, we used a higher oxygen flow rate and plasma power from the following paper [Huang et al., A ground-based radio frequency inductively coupled plasma apparatus for atomic oxygen simulation in low Earth orbit. Review of Scientific Instruments 2007, 78, 103301], while keeping all other parameters the same. Since they have calculated and shown that the effective AO flux is 100 times more than that in the LEO environment, we expect that our effective AO flux to be the same or higher. Hence the AO environment simulated here is also appropriate.

These additional information and discussion points are now incorporated into the characterisation methods section now, as per the Reviewer’s suggestion. The relevant revisions can be found on page 3 of the revised manuscript.  

  1. Did the Authors make mapping of the structure (e.g. Raman) before and after the exposition?

Thank you for the suggestion, following the advice, we have provided a Raman spectroscopy of before and after AO exposure. There was no shift in the characteristic peak. This is similar to what other groups have observed for thermal oxidation: that bulk BN is much more resistant to oxidation that few layered BN [see e.g., Ref. Li et al., Strong oxidation resistance of atomically thin boron nitride nanosheets. 291 ACS nano 2014, 8, 1457–1462, which is cited as Ref 33 in the revised manuscript]. The relevant revisions can be found on page 9 of the revised manuscript.  

  1. Did the Authors repeated the experiment and got the same results?

Yes, the experiments are repeatable. For the etching studies of Xe and AO, samples grown from different CVD runs were randomly selected. The etching rate was confirmed to be reproducible and mentioned on page 3 of the revised manuscript.

  1. Section "Conclusions" is too short and too general. It was not confirmed by numerical parameters that success was achieved.

We thank the Reviewer for this important suggestion. The conclusion has been rewritten following the recommended points such as the specific achievements and the numerical parameters (see pages 11 of the revised manuscript). This has made the paper significantly better. 

Round 2

Reviewer 1 Report

The authors corrected major issues and sufficiently improved the manuscript.

Anyway, the manuscript requires some minor changes and additions.

1.    Despite the citation of previous contributions, the XPS spectra should be provided for all materials before and after treatment.

2.    Please, follow the principles of IMRAD format for the preparation of manuscript with isolated experimental, results and discussion section, conclusion.

Author Response

Reviewer 1 comments:

Anyway, the manuscript requires some minor changes and additions.

  1. Despite the citation of previous contributions, the XPS spectra should be provided for all materials before and after treatment.

Thank you reviewer for your comments. We agree with your concerns and have thus provided the XPS analysis to indicate the surface oxidation states of the BNNS before and after AO exposure. For your convenience, the relevant graphs and discussion is attached below:

(Please see attached document for picture, unable to upload picture into comments.)

The surface oxidation states of BNNS thin film before and after AO exposure was also characterised using XPS. A clear B-N and N-B signal at 190.5eV and 398.2eV can be detected from the B1s and N1s spectra (8a-b) before exposure to the AO[ 35]. After 40 hours of exposure, B-O bonds with a characteristic peak of 193.5eV is found[ 36 ]. In the N1s spectra, two peaks are shown at 398.9eV and 402.0eV, which corresponds to the N-C and N-O bonds respectively [ 37 ]. The formation of the N-C bonds is likely due to the interaction of adventitious carbon with the nitrogen of the BN under the plasma of our simulated AO environment. From the XPS and Raman data, it is evident that the AO indeed interacted with the BNNS thin film. However, as the deposited BNNS is 300nm, only the surface interacts with the AO, leaving most of the BNNS intact as BN. Therefore, B-O and N-O bonds can be observed on the surface via XPS, but the BN remains largely intact as shown by the Raman spectroscopy in Figure 7.

  1. Please, follow the principles of IMRAD format for the preparation of manuscript with isolated experimental, results and discussion section, conclusion.

Thank you for pointing that out. The paper is now divided into 4 parts: Introduction (from page 1), Methods (from page 2), Results and Discussion (from page 4) and lastly Conclusion (from page 10).

Reviewer 2 Report

Paper fits the scope of the journal although its novelty is moderate. Homogeneity and durability of the layers could be better confirmed. However, after improvement provide by Authors, it can be considered for publication.

Author Response

Reviewer 2 comments:

Paper fits the scope of the journal although its novelty is moderate. Homogeneity and durability of the layers could be better confirmed. However, after improvement provide by Authors, it can be considered for publication.

Thank you reviewer 2 for considering this paper for publication.